# *IL-10* Gene Rs1800871, Rs1800872, and Rs1800896 Polymorphisms and IL-10 Serum Levels Association with Pituitary Adenoma

**DOI:** 10.3390/biomedicines10081921

**Published:** 2022-08-08

**Authors:** Migle Palivonaite, Greta Gedvilaite, Brigita Glebauskiene, Loresa Kriauciuniene, Vita Rovite, Rasa Liutkeviciene

**Affiliations:** 1Neuroscience Institute, Medical Academy, Lithuanian University of Health Sciences, Eiveniu 2, 50161 Kaunas, Lithuania; 2Medical Academy, Lithuanian University of Health Sciences, Eiveniu 2, 50161 Kaunas, Lithuania; 3Latvian Biomedical Research and Study Centre (BMC), LV-1046 Riga, Latvia

**Keywords:** pituitary adenoma, *IL-10* gene polymorphisms, IL-10 serum levels

## Abstract

The aim and objective of this study is to determine the association between the rs1800871, rs1800872, and rs1800896 polymorphisms of the gene *IL-10* and the serum levels of IL-10 in patients with pituitary adenoma. Methods: Data from 106 patients with pituitary adenoma and 192 control patients were used for the study. DNA was isolated from peripheral blood using the salt precipitation method. The samples were genotyped in real-time using the polymerase chain reaction method. IL-10 serum levels were evaluated using an ELISA kit. The data obtained were systematized using the computer program IBM SPSS Statistics. Results: The AG genotype of *IL-10* rs1800871 was statistically significantly lower in the inactive PA group than in the control group (22.7% vs. 40.6%, *p* = 0.027). The TG genotype of *IL-10* rs1800872 was also statistically significantly lower in the inactive PA group than in the control group (22.7% vs. 40.6%, *p* = 0.027). A binary logistic regression analysis of the polymorphisms in the *IL-10* gene in PA and control groups based on the pituitary adenoma activity showed that the AG genotype of *IL-10* rs1800871 increased the chance of inactive PA by 2.2-fold in codominant (OR: 2.272, CI: 1.048–4.925, *p* = 0.038) and overdominant (OR: 2.326, CI: 1.086–4.982, *p* = 0.030) models. Moreover, the TG genotype of *IL-10* rs1800872 increased the probability of inactive PA by 2.2-fold in codominant (OR: 2.272, CI: 1.048–4.925, *p* = 0.038) and overdominant (OR: 2.326, CI: 1.086–4.982, *p* = 0.030) models. The association of the *IL-10* polymorphisms with PA invasiveness and recurrence in PA and control groups did not yield statistically significant results. Conclusions: *IL-10* rs1800871 and *IL-10* rs1800872 may be associated with the development of inactive PA.

## 1. Introduction

Pituitary adenomas (PA) are described as benign monoclonal tumors in the pituitary gland that cause hormone hypersecretion or localization symptoms. PA are primary tumors that occur in the pituitary gland and are among the most common intracranial tumors, with a prevalence of approximately 0.1 percent in the population [1]. Pituitary tumors can occur in people of any age, including children, but are more common in the elderly [2].

The pathogenesis of pituitary adenomas is still poorly understood. Although many consider benign monoclonal proliferation, their clinical spectrum is diverse, including hormone hypersecretion and various degrees of invasiveness, which suggests multiple steps and mechanisms as well as molecular abnormalities in tumorigenesis and gene regulation of pituitary adenomas [3].

The human *IL-10* gene is located on chromosome 1q31–32 (10), a locus genetically associated with susceptibility to a number of autoimmune diseases [4]. Several polymorphisms have been identified within the *IL-10* locus, including 23 single nucleotide polymorphisms (SNPs) localized to the promoter alone. Although most SNPs are in linkage disequilibrium, it is reasonable to assume that some may play a biological role in regulating the IL-10 expression [5].

IL-10 was initially considered as a T helper-2 cytokine that modulates the growth and differentiation of innate immune cells, endothelial cells, and keratinocytes and suppresses the activation and functions of T cells [6]. Numerous types of cells can produce IL-10, such as B cells, macrophages, mast cells, dendritic cells, and epithelial cells in mice and humans [7,8]. IL-10 has been considered a suppressive factor affecting proliferation, cytokine production, and migration of effector T cells [9]. It is known that elevated levels of IL-10 inhibited cytolytic activity in transplanted tumors [10]. IL-10 administration suppressed tumor growth and resulted in tumor rejection in several tumors, including melanoma, sarcoma, and colorectal cancer [11,12,13,14]. According to other studies, IL-10 may also promote the development of glioma, lung cancer, squamous cell carcinoma of the esophagus, melanoma, and cervical cancer [15,16,17,18,19].

As early as 1995, studies showed that expressed interleukin-10 (IL-10) affects pituitary tumor cells of mice and isolated mouse pituitary glands [20]. Interleukin-10 has a wide range of activities in immune and neuroendocrine interactions [21]. In particular, it may have the potential to mimic some of the activities of ACTH in the immune system, based on their shared ability to inhibit interferon-gamma (IFN-y) production [22] and modulate B lymphocyte responses [23]. These results provide a good basis for further defining the role of IL-10 in the immune and neuroendocrine axes. The data of Hughes et al. show production of IL-10 by pituitary cells, induction of ACTH by IL-10 in pituitary cells, modulation of ACTH production in lymphocytes by IL-10, and identification of IL-10 mRNA in pituitary cells by reverse transcriptase-coupled polymerase chain reaction (PCR) followed by a sequence analysis. These data suggest that IL-10 plays an important role in the interaction between the immune system and the neuroendocrine system [24].

Huang et al. state that IL-10 may be a helpful biomarker for diagnosing invasive, non-functioning PA. Huang’s team compared IL-10 expression in patients with non-invasive, non-functioning PA with that of patients with invasive, non-functional PA and found significantly higher interleukin-10 expression. These results suggest that high IL-10 expression in pituitary tumor tissue is associated with increased invasiveness of non-functioning PA, so IL-10 levels in peripheral blood can be used as a diagnostic marker for invasive non-functional PA [25].

Therefore, this study aims to investigate the polymorphisms of the *IL-10* gene, rs1800871, rs1800872, and rs1800896, and IL-10 serum levels in association with pituitary adenomas.

## 2. Materials and Methods

Permission (No. BE-2-47) to conduct the study was granted by the Ethics Committee for Biomedical Research.

A PA group and a healthy control group were included in the study. The inclusion criteria for the PA group were: PA, diagnosis, and confirmation by magnetic resonance imaging (MRI), good general health, informed consent to participate in the study, age over 18 years, and absence of other tumors.

The group of healthy controls was composed according to the distribution of sex and age in the pituitary adenoma group.

### 2.1. DNA Extraction and Genotyping

After venous blood samples (white blood cells) were collected, the DNA salting-out method was used to prepare genomic DNA. The method is based on the collection of cells by centrifugation and their suspension in a buffer solution, degradation of cell membranes with detergents, hydrolysis of proteins by proteinase K and deproteinization with chloroform, and precipitation of the DNA with ethanol.

Genotyping of *IL-10* rs1800871, rs1800872, and rs1800896 was performed using a real-time PCR (polymerase chain reaction). Genotyping of the three SNPs was performed using the Step One Plus real-time PCR system (Applied Biosystems, Chicago, IL, USA). TaqMan^®^ SNP genotyping assays (Thermo Scientific, Waltham, MA, USA) for all SNPs were performed according to the manufacturer’s protocol. During the real-time PCR, the Allelic Discrimination Program was used. This program determined the individual genotypes based on the fluorescence intensity of the different detectors (VIC and FAM). Validated assays were used for the real-time PCR: rs1800871 Context Sequence [VIC/FAM]: AGTGAGCAAACTGAGGCACAGAGAT[A/G]TTACATCACCTGTACAAGGGTACAC; rs1800872 Context Sequence [VIC/FAM]: CTTTCCAGACTGGCTTCCTACAG[T/G]ACAGGCGGTCACAGGATGTTC; and rs1800896 Context Sequence [VIC/FAM]: TCCTCTTACCTATCCCTACTTCCCC[T/C]TCCCAAAGAAGCCTTAGTAGTGTTG.

PA invasiveness, activity, recurrence assessment, control group formation, DNA extraction, and genotyping have been described previously in our studies [26,27].

### 2.2. IL-10 Protein Measurement

IL-10 serum levels were measured in 41 control subjects and 28 patients with PA. The assay was determined by enzyme-linked immunosorbent assay (ELISA) using the Invitrogen IL-10 Human ELISA Kit. This assay for human IL-10, with a standard curve sensibility range of 0.9–500 pg/mL and a sensitivity of <1 pg/mL was analyzed on the Multiskan FC Microplate Photometer (Thermo Scientific, Waltham, MA, USA) at 450 nm per the manufacturer’s instructions. The samples were excluded if the levels of serum cytokines were below the detection range.

### 2.3. Statistical Analysis

We calculated the deviation from HWE of the *IL-10* gene rs1800871, rs1800872, and rs1800896 polymorphisms in the present report. Pearson’s χ2 statistical test was used in both the case and control subjects. The data were analyzed using SPSS 27.0 statistical analysis software (IBM SPSS, Armonk, NY, USA). The frequencies of the *IL-10* rs1800871, rs1800872, and rs1800896 genotypes and alleles between the case and control subjects were compared using the χ2 test in all groups. The nonparametric Mann–Whitney U test compared nonnormally distributed continuous data, and Pearson’s χ2 test was used to compare categorical variables. The genotype-based odds ratios (OR) and 95% confidence intervals (Cis) were estimated using binary logistic regression models regarding the effects of genotypes on the development of PA. The data are presented as absolute numbers with percentages in parentheses, a median, and interquartile range (IQR). A *p*-value of less than 0.05 indicated a statistically significant difference.

The *IL-10* haplotype association analysis was performed separately in the patients with pituitary adenoma and control groups. We used the online SNPStats website (https://www.snpstats.net/snpstats/) (accessed on 12 March 2022). The pairwise linkage disequilibrium (LD) analysis was assessed by D’ and r2 measures. The associations between the haplotypes and multiple sclerosis were calculated by logistic regression and presented as ORs and 95% CI.

## 3. Results

The patients with pituitary adenoma consisted of 106 subjects, of whom 60 (56.6%) were women, and 46 (43.3%) were men. The subjects ranged from 18 to 84 years, with a mean age of 53.4 years. The control group consisted of 192 subjects, of whom 129 (67.2%) were women, and 63 (32.8%) were men. The age of the control group ranged from 18 to 94 years, with a mean age of 55.8 years. The sex and age did not differ between the groups (*p* = 0.069 and *p* = 0.199, respectively) (Table 1).

### 3.1. Associations of IL-10 rs1800871, rs1800872, and rs1800896 with Pituitary Adenoma

We compared the distribution of genotypes and alleles’ frequencies between the PA and control groups and found no statistically significant differences (Appendix A). The binary logistic regression analysis also did not yield statistically significant results (Appendix A). Analyses was performed separately for males and females, but no statistically significant results were found in either the distribution of genotypes/alleles or the binary logistic regression (Appendix A).

### 3.2. Associations of IL-10 rs1800871, rs1800872, and rs1800896 with Pituitary Adenoma’s Invasiveness

The adenomas’ invasiveness distribution of the genotypes and alleles’ analysis was performed between the non-invasive PA group and the control group, as well as between the invasive PA group and the control group. However, the distribution analysis and binary logistic regression analysis did not reveal statistically significant results (Appendix A).

### 3.3. Associations of IL-10 rs1800871, rs1800872, and rs1800896 with Pituitary Adenomas’ Activity

Pituitary adenoma was divided into active and inactive groups. We found that the IL-10 rs1800871 AG genotype was statistically significantly less frequent in the inactive PA group than in the control group (22.7% vs. 40.6%, *p* = 0.027), and the rs1800872 TG genotype was statistically significantly less frequent in the inactive PA group than in the control group (22.7% vs. 40.6%, *p* = 0.027) (Table 2).

A binary logistic regression analysis was performed to evaluate the impact of *IL-10* gene rs1800871, rs1800872, and rs1800896 on active/inactive PA development. The analysis revealed that *IL-10* rs1800871 AG genotype increased the odds of inactive PA’s development 2.2.-fold under the codominant (OR: 2.272, CI: 1.048–4.925, *p* = 0.038) and overdominant (OR: 2.326, CI: 1.086–4.982, *p* = 0.030) models. Moreover, the IL-10 rs1800872 TG genotype increases the odds of inactive PA development 2.2-fold under the codominant (OR: 2.272, CI: 1.048–4.925, *p* = 0.038) and overdominant (OR: 2.326, CI: 1.086–4.982, *p* = 0.030) models (Table 3).

### 3.4. Associations of IL-10 rs1800871, rs1800872, and rs1800896 with Pituitary Adenomas’ Recurrence

When comparing the distribution of the genotypes and alleles between the PA with the recurrence group vs. the control group and the PA without the recurrence group vs. the control group, no statistically significant differences were found (Appendix A); the binary logistic regression analysis also did not reveal statistically significant results (Appendix A).

### 3.5. Serum IL-10 Levels

IL-10 serum levels were measured in 41 control group subjects and in 28 PA patients. The analysis showed lower IL-10 serum levels in the PA group compared to the control subjects (median (IQR): 0.328 (1.015) pg/mL vs. 5.9 (8.453) pg/mL, *p* < 0.001) (Appendix A). A comparison between the serum IL-10 levels of the study groups and genotypes was performed. The IL-10 levels were statistically significantly lower in the PA than in the control group with the rs1800871 GG and GA genotypes (median (IQR): 0.373 (1.023) vs. 0.924 (8.404), *p* = 0.018; 0.392 (1.665) vs. 8.474 (11.784), *p* = 0.016, accordingly). We noticed exactly the same results with rs1800872: the IL-10 levels were lower in the PA than in the control group with the GG and GT genotypes (median (IQR): 0.373 (1.023) vs. 0.924 (8.404), *p* = 0.018; 0.392 (1.665) vs. 8.474 (11.784), *p* = 0.016, accordingly). The IL-10 levels were also found to be statistically significantly lower in the PA than in the controls with the rs1800896 TT (median (IQR): 0.642 (1.634) vs. 7.796 (8.782), *p* = 0.007) and TC genotypes (median (IQR): 0.231 (4.430) vs. 8.300 (32.670), *p* = 0.010) (Table 4).

### 3.6. IL-10 rs1800871, rs1800872, rs1800896 Haplotype Analysis

Haplotype analysis was performed in the PA and the control groups. A pairwise linkage disequilibrium (LD) between rs1800871–rs1800872, rs1800871–rs1800896, and rs1800872–rs1800896 was observed (results are shown in Table 5).

However, the haplotype frequency analysis did not show any associations with PA (Appendix A).

## 4. Discussion

The study analyzed the associations of the *IL-10* (rs1800871, rs1800872, and rs180089) gene polymorphisms and IL-10 serum levels in patients with PA and healthy controls. The results were evaluated according to gender and the clinical course of the disease. To our knowledge, the associations of the *IL-10* gene polymorphisms (rs1800871, rs1800872, and rs1800896) and IL-10 serum levels with PA were analyzed for the first time.

IL-10 is associated with many autoimmune diseases due to its anti-inflammatory functions. IL-10 inhibits innate and adaptive immune responses of leukocytes and limits potential tissue damage caused by inflammation. IL-10 is essential for host defense against various infections and developing many autoimmune diseases [28]. However, IL-10 may play a dual role in tumorigenesis and development. When cancer forms, IL-10 may mainly stimulate NK- and CTL-mediated killing of cancer cells. If cancer cells survive and rewire to express IL-10 production in the tumor microenvironment, it may mainly act as a potent cancer promoter [29]. Investigations have shown that the IL-10 promoter region polymorphisms affect *IL-10*’s gene transcription and translation, resulting in abnormal cell proliferation and cancer development [30].

Only one study has analyzed *IL-10* in association with PA. Huang and co-authors associated the gene expression of *IL-10* with PA. The PRL (µg/L) levels were significantly higher, but the LH and FSH levels were lower in patients with non-invasive and invasive, non-functional PA compared to the control group. Furthermore, patients with invasive, non-functional PA had significantly lower white blood cell counts than patients with non-invasive, non-functional PA and the control group. The study showed that a high IL-10 expression in pituitary tumor tissues was associated with an increased invasiveness of non-functioning PA; thus, IL-10 levels in peripheral blood can be used as a diagnostic marker for invasive non-functioning PA [25]. In our study, *IL-10* rs1800871 and rs1800872 were associated with PA. The results of the study showed a statistically significant difference when comparing the inactive PA group with the control group. *IL-10* rs1800871 AG and rs1800872 TG genotypes were less frequent in the inactive PA group than in the control group (*p* = 0.027). Depending on the activity of the PA, individuals with the *IL-10* rs1800871 AG genotype or rs1800872 TG genotype were 2.2 times more likely to be an inactive PA in codominant and overdominant models. The other group of authors also studied patients with nonfunctioning but invasive PA (NFPAs) and found lower numbers of CD3-CD56 + natural killer (NK) cells and significantly higher levels of CD3 + CD8 + CD28 T cells (CD8 + Tregs) and interleukin-10 (IL-10) in peripheral blood. In addition, patients with invasive NFPAs had fewer infiltrated CD56 + cells, fewer infiltrated CD28 + cells, and significantly higher IL-10 expression. These results indicated that low infiltration of CD56 + cells and CD28 + cells and a high IL-10 expression in pituitary tumor tissues were related to an increased invasiveness of NFPAs. The levels of CD3-CD56 + NK cells, CD8 + Tregs, and IL-10 in peripheral blood may be useful as diagnostic markers for invasive NFPAs [25].

Numerous studies have been conducted to link the gene *IL-10* to other types of brain tumors, cancer, and CNS diseases. Several authors have linked the gene *IL-10* to glioma cell proliferation. Zhang et al. link the *IL-10* gene to glioma cell proliferation. The study showed that IL-10 increased cell growth and invasion into glioma cells. IL-10 has been shown to enhance the expression of KPNA2, which affects cell growth and expression in cancer cells, including the human brain tumor, suggesting that IL-10 promotes glioma cell growth and invasion by regulating KPNA2 [31]. Another study reported that the IL-10 rs1800869 GG genotype causes an increase in IL-10 production, suggesting that the mutant allele G has significantly protective associations against glioma [32]. Geng and other authors have shown that the *IL-10* gene is involved in the development of lymphoid malignancies. The study’s results showed that the level of IL-10 in the cerebrospinal fluid correlates positively with the tumor size and may help diagnose the development of lymphoid malignancies [33]. Lissoni et al., in a study of people with various localized cancers, found that the IL-10 levels were severely high in 14 of 50 (28%) cancer patients and that the IL-10 levels were higher in patients with hypercortisolemia than in those with normal cortisol levels [34]. Rubenstein and co-authors linked the *IL-10* gene to primary and secondary CNS lymphomas. Concentrations of IL-10 in the cerebrospinal fluid in primary or secondary CNS lymphoma were also significantly higher than in samples from healthy subjects [35].

Numerous studies have investigated the association between the polymorphisms of *IL-10* as potential biomarkers in different types of cancer. Our study results showed that *IL-10* rs1800871 and rs1800872 were associated with PA. Moreover, Wang and co-authors reported that the *IL-10* rs1800871, rs1800872, and rs1800896 polymorphisms were significantly associated with the risk of gastric cancer in Asians under two models: dominant and additive. In this case, our study indicated that the *IL-10* polymorphisms (rs1800871, rs1800872, rs1800896) could serve as genetic biomarkers of gastric cancer in Asians [36]. Moghimi and other authors found that there was a significant association between the IL-10 rs1800872 polymorphism and risk of breast cancer under four genetic models: allele, homozygote, dominant, and recessive [37]. Makni and co-authors analyzed the *IL-10* polymorphisms’ susceptibility to head and neck cancers (HNC). This study reported that the rs1800871 C/C genotype was associated with a 2.5-fold and 3.33-fold risk of developing nasopharyngeal cancer (NPC) and laryngeal cancer (LC), respectively, and carriers of the rs1800872 A/A genotype had a 2.5-fold increased risk of HNC and NPC, but not LC. The investigation findings showed that the *IL-10* rs1800896, rs1800871, and rs1800872 SNPs contribute to the development of NPC, which suggests a possible role for these variants as biomarkers for the early detection of HNC and, in particular, the NPC subtype [38]. In the study conducted by Yang et al., *IL-10* (rs1800896), the AA genotype and GA + AA genotype were associated with an enhanced risk of esophageal cancer; thus, this variant may be performed as a candidate biomarker for predicting the development of esophageal cancer [39].

Our study found that the IL-10 levels were statistically significantly lower in the PA than in the control group with the rs1800871, rs1800872 GG and GA genotypes, and with the rs1800896 TT genotype. Several authors reported that IL-10 serum levels were important in cancer development. Bobe et al. revealed that elevated serum levels of IL-10 increased the risk of colorectal adenoma recurrence [40]. Shokrzadeh and other authors reported that the serum IL-10 levels were significantly higher in the patients with gastric adenocarcinoma than in the healthy controls. Elevated levels of IL-10 may be useful as diagnostic biomarkers for this adenocarcinoma [41]. Feng and co-authors analyzed the serum IL-10 levels in patients with pancreatic cancer. Their study showed that patients with high IL-10 expressions had a significantly shorter survival time than patients with low expressions. The study found that a high expression of IL-10 was strongly associated with poor survival [42]. Therefore, a study by Guo et al. suggest that IL-10 rs1800871 may be a useful biomarker for predicting glioma patient outcomes [43]. Another study showed that patients with primary central nervous system lymphomas (PCNSLs) and primary vitreoretinal lymphomas (PVRLs) had similar IL-10 (-1082) A allele frequencies, but their genotype distributions differed from those of the healthy controls. The findings suggest that the IL-10 (-1082) A allele is a risk factor for higher IL-10 levels in PVRLs and PCNSLs. Higher IL-10 levels have been correlated with more aggressive disease in both PVRLs and PCNSLs, making this finding an important and potentially clinically significant observation [44]. Indeed, there is only one study analyzing another IL-9 SNP, conducted by Mickevicius et al. This study proved that the G/G genotype and G allele of the *IL-9* (rs1859430) polymorphism are more prevalent among patients with a recurrent PA. Moreover, our results show the that A/A genotype and A allele are significantly more common among those patients for whom the PA did not reoccur. We can hypothesize that IL-9 may be important for PA recurrence, but it has no impact on hormonal activity or invasiveness [45].

Based on the results of studies that investigated the association between the gene *IL-10* and the development of brain tumors, CNS alterations, and other types of cancers, it is possible that changes in gene expression influence disease.

Currently, there are insufficient studies on the *IL-10* gene in CNS diseases and brain tumors, especially PA. More research and an increase in sample size would provide more information on the effects of this gene on pituitary tumor development, progression, and activity.

Personalized medicine plays the most important role in the prevention, diagnosis, prognosis, and therapy of neoplasms and cancer. It is important for clinical management in routine clinical practice and treatment. Identification of genes that are predisposed to neoplasms and cancer can help make decisions for an individual’s disease risk modification. The study of genes in different populations is very valuable because the prevalence of genes varies widely in very different ethnic groups. In the future, we need to focus on the role of personalized medicine, including the most important: one-gene therapy medicine.

## Figures and Tables

**Table 1 biomedicines-10-01921-t001:** Demographics.

Characteristics	Group		*p*-Value
	I Group: PA (*n* = 106)	II Group: Control (*n* = 192)	
Females	60 (56.6%)	129 (67.2%)	0.069 *
Males	46 (43.3%)	63 (32.8%)	
Age, mean (SD)	53.4 (15.5)	55.8 (15.4)	0.199 **
Invasiveness	105 (99.1%)	-	-
Data not known	1 (0.9%)		
Activity	102 (96.2%)	-	-
Data not known	4 (3.8%)		
Recurrence	102 (96.2%)	-	-
Data not known	4 (3.8%)		

* Pearson chi-square was used; ** Student’s *t*-test was used.

**Table 2 biomedicines-10-01921-t002:** Distributions of *IL-10* rs1800871, rs1800872, rs1800896 genotypes and alleles in patients with PA and control groups by PA activity.

Gene	Genotype	Control (n = 192)	Inactive PA	*p*-Value	Active PA (n = 58) *n* (%)	*p*-Value
		*n* (%)	(n = 44) *n* (%)			
*IL-10* (rs1800871)	AA	11 (5.7)	4 (9.1)	0.08	3 (5.2)	0.8
	AG	**78 (40.6) ***	**10 (22.7) ***		21 (36.2)	
	GG	103 (53.6)	30 (68.2)		34 (58.6)	
	Allele:			0.275		0.549
	A	100 (26.0)	18 (20.5)		27 (23.3)	
	G	284 (74.0)	70 (79.5)		89 (76.7)	
*IL-10* (rs1800872)	GG	103 (53.6)	30 (68.2)	0.08	35 (60.3)	0.665
	TG	**78 (40.6) ****	**10 (22.7) ****		20 (34.5)	
	TT	11 (5.7)	4 (9.1)		3 (5.2)	
	Allele:			0.275		0.43
	G	284 (74.0)	70 (79.5)		90 (77.6)	
	T	100 (26.0)	18 (20.5)		26 (22.4)	
*IL-10* (rs1800896)	CC	43 (22.4)	11 (25.0)	0.869	12 (20.7)	0.844
	TC	100 (52.1)	21 (47.7)		29 (50.0)	
	TT	49 (25.5)	12 (27.3)		17 (29.3)	
	Allele:			0.942		0.604
	C	186 (48.4)	43 (48.9)		53 (45.7)	
	T	198 (51.6)	45 (51.1)		63 (54.3)	

* AG vs. AA + GG, *p* = 0.027; ** TG vs. GG + TT, *p* = 0.027.

**Table 3 biomedicines-10-01921-t003:** Binary logistic regression analysis of *IL-10* rs1800871, rs1800872, and rs1800896 in the PA and control groups by PA activity.

Model	Genotype/Allele	OR (95% CI)	*p*-Value	AIC
*IL-10* (rs1800871)				
Inactive PA				
Codominant	AG vs. GG	2.272 (1.048–4.925)	**0.038**	225.717
	AA vs. GG	0.801 (0.238–2.698)	0.720	
Dominant	AG + AA vs. GG	1.852 (0.924–3.710)	0.082	225.89
Recessive	AA vs. GG + AG	0.608 (0.184–2.007)	0.414	228.419
Overdominant	AG vs. AA + GG	2.326 (1.086–4.982)	**0.030**	223.842
Additive	G	1.368 (0.776–2.410)	0.278	227.813
Active PA				
Codominant	AG vs. GG	1.226 (0.661–2.276)	0.518	274.394
	AA vs. GG	1.210 (0.319–4.595)	0.779	
Dominant	AG + AA vs. GG	1.224 (0.675–2.218)	0.505	272.394
Recessive	AA vs. GG + AG	1.114 (0.300–4.137)	0.872	272.814
Overdominant	AG vs. AA + GG	1.206 (0.656–2.214)	0.547	272.475
Additive	G	1.169 (0.710–1.924)	0.540	272.459
*IL-10* (rs1800872)				
Inactive PA				
Codominant	TG vs. GG	2.272 (1.048–4.925)	**0.038**	223.717
	TT vs. GG	0.801 (0.238–2.698)	0.720	
Dominant	TG + TT vs. GG	1.852 (0.924–3.710)	0.082	225.89
Recessive	TT vs. GG + TG	0.608 (0.184–2.007)	0.414	228.419
Overdominant	TG vs. GG + TT	2.326 (1.086–4.982)	**0.030**	223.842
Additive	G	1.368 (0.776–2.410)	0.278	227.813
Active PA				
Codominant	TG vs. GG	1.325 (0.711–2.471)	0.376	274.019
	TT vs. GG	1.246 (0.329–4.725)	0.746	
Dominant	TG + TT vs. GG	1.315 (0.723–2.390)	0.369	272.027
Recessive	TT vs. GG + TG	1.114 (0.300–4.137)	0.872	272.814
Overdominant	TG vs. GG + TT	1.300 (0.704–2.400)	0.402	272.128
Additive	G	1.229 (0.743–2.032)	0.422	272.181
*IL-10* (rs1800896)				
Inactive PA				
Codominant	TC vs. TT	0.858 (0.390–1.884)	0.702	230.761
	CC vs. TT	0.821 (0.364–1.850)	0.634	
Dominant	TC + CC vs. TT	0.840 (0.436–1.618)	0.602	228.77
Recessive	CC vs. TT + TC	0.866 (0.404–1.855)	0.711	228.906
Overdominant	TC vs. TT + CC	0.914 (0.437–1.912)	0.811	228.985
Additive	T	0.901 (0.605–1.342)	0.609	228.782
Active PA				
Codominant	TC vs. TT	0.836 (0.420–1.665)	0.610	274.507
	CC vs. TT	1.039 (0.485–2.226)	0.921	
Dominant	TC + CC vs. TT	0.920 (0.511–1.655)	0.781	272.763
Recessive	CC vs. TT + TC	1.106 (0.538–2.273)	0.783	272.765
Overdominant	TC vs. TT + CC	0.826 (0.431–1.586)	0.566	272.517
Additive	T	0.994 (0.691–1.431)	0.975	272.84

OR: odds ratio; CI: confidence interval; AIC: Akaike information criteria; *p*-value: significance level (statistically significant when *p* < 0.05).

**Table 4 biomedicines-10-01921-t004:** Genotype distribution and serum IL-10 levels.

Gene, Genotype	IL-10 Level (pg/mL)		*p*-Value
	PA Median (IQR)	Controls Median (IQR)	
rs1800871			
GG	0.373 (1.023)	0.924 (8.404)	0.018 *
GA	0.392 (1.665)	8.474 (11.784)	0.016 **
AA	0.170 (-)	32.712 (-)	0.083 **
rs1800872			
GG	0.373 (1.023)	0.924 (8.404)	0.018 *
GT	0.392 (1.665)	8.474 (11.784)	0.016 **
TT	0.171 (-)	32.712 (-)	0.083 **
rs1800896			
TT	0.642 (1.634)	7.796 (8.782)	0.007 *
TC	0.231 (4.430)	8.300 (32.670)	0.010 *
CC	0.207 (0.177)	0.709 (2.224)	0.174 **

* Mann–Whitney test; ** Student’s *t*-test.

**Table 5 biomedicines-10-01921-t005:** Linkage disequilibrium between every two *IL-10* SNPs.

SNPs	PA vs. Control Groups		
	D’	r^2^	*p*-Value
rs1800871–rs1800872	0.9996	0.9902	0.0
rs1800871–rs1800896	0.9802	0.2896	0.0
rs1800872–rs1800896	0.9801	0.2898	0.0

D’: deviation between the expected haplotype frequency and the observed frequency [D’ scale: 0,1]; r^2^: squared correlation coefficient of the haplotype frequencies [r^2^ scale: 0,1]; *p*: significance level when *p* = 0.05.

## Data Availability

The data used to support the findings of this study are available from the corresponding author upon request.

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
