# Peer review of "IL-10 Gene Rs1800871, Rs1800872, and Rs1800896 Polymorphisms and IL-10 Serum Levels Association with Pituitary Adenoma"

_biomedicines, 2022, doi:10.3390/biomedicines10081921_

Round 1

Reviewer 1 Report

The article offers novel insights into the development of pituitary adenoma (PA). The authors have proven that the IL-10 rs1800872 TG genotype and IL-10 rs1800871 AG genotype could have a protective role against developing inactive PA. The study design is well organized and easy to follow.

The article, however, has some major issues:

-        In my opinion, there are not enough novel results for a full original article. I think that the study and the results would fit much better in a short communication.

-        There is a lot of repetition of literature data from the introduction, in the discussion section. The information gets redundant many times.

-        In the introduction, the authors should make clearer as to why did they choose to look at IL-10 specifically

-        In the discussion section, while it would be useful to look at how IL-10 polymorphism and level affects other types of cerebral tumors, it is too far stretched to describe its role in breast cancer, gastric cancer or head and neck cancer. Instead, the authors could add details about how polymorphisms in other interleukins/immune related genes are linked to PA

-        The methods and materials section is far too short. Even if some methods have been previously described in other studies, some description should still be provided for each of the used methods. The methods and materials section should be divided into subchapters. There are some details missing. For instance, which ELISA kit was used?

-        The results section should also be divided into smaller subchapters, each one containing at the end a short summary of the findings

-        It could be interesting and more engaging to replace the way the results are presented. There are too many tables, and it gets difficult to follow the information. For instance, the ELISA results could be represented as graphs

-        More details about inactive PA should be given, since the discovered polymorphisms are linked specifically to this type of PA

-        At the end of the discussion section, more details about how the results could be further analyzed or what it can mean for the study of PA should be given. For instance, the authors can include talks about personalized medicine and patients’ stratification or risk assessment in the general population

Author Response

The article offers novel insights into the development of pituitary adenoma (PA). The authors have proven that the IL-10 rs1800872 TG genotype and IL-10 rs1800871 AG genotype could have a protective role against developing inactive PA. The study design is well organized and easy to follow.

Thank you for your comment. We have made some revisions in response to your comments

The article, however, has some major issues:

-        In my opinion, there are not enough novel results for a full original article. I think that the study and the results would fit much better in a short communication.

Thank you for your comment. Besides the fact that rs1800871, rs1800872, and rs1800896 polymorphisms were investigated with several other cancer types, its important to mention, that this is the first study that investigates these SNPs associations with pituitary adenoma.

From the scientific point of view a short communication paper may be used if you are working with a hot topic and had discovered something never explored before in the literature.

This is done to ensure that you are the first scientist to report this feature. In some research topic thing go very fast and in some weeks another author can publish what you have discovered!

Normally a communication need to be about a really novel feature!

-        There is a lot of repetition of literature data from the introduction, in the discussion section. The information gets redundant many times.

Thank you for your comment. The manuscript was revised.

-        In the introduction, the authors should make clearer as to why did they choose to look at IL-10 specifically.

Thank you for your comment. The information was added into the introduction section.

Interleukin-10 has a wide range of activities in immune and neuroendocrine interactions (Howard, M., O'Garra, A., lshida, H., de Waal Malefyt, R., and de Vries, J. (1992). Biological properties of IL-10. J. Clin. lmmunoL 12.'239-247.). In particular, it may have the potential to mimic some of the activities of ACTH in the immune system, based on their shared ability to inhibit interferon-gamma (IFN-y) production (Fiorentino, D. F., Bond, M. A., and Mosmann, T. R. (1989). Two types of mouse helper T-cells IV. Th2 clones secrete a factor that inhibits cytokine production by Thl ceils. J. Exp. Med. 170".2081-2095) and modulate B lymphocyte responses (Rousset, F., Garcia, E., DeFrance, T. E., Peronne, C., Vezzio, N., Hsu, D., Kastelein, R., Moore, K. W., and Bancherau, J. (1992). Intedeukin-10 is a potent growth and differentiation factor for activated human B-lymphocytes. Proc. Natl. Acad. Sci. USA 89:t890-1893.). These results provide a good basis for further defining the role of IL -10 in the immune and neuroendocrine axes. The data of Hughes et. Al. show production of IL -10 by pituitary cells, induction of ACTH by IL -10 in pituitary cells, modulation of ACTH production in lymphocytes by IL -10, and identification of IL -10 mRNA in pituitary cells by reverse transcriptase-coupled polymerase chain reaction (PCR) followed by sequence analysis. These data suggest that IL -10 plays an important role in the interaction between the immune system and the neuroendocrine system (Thomas K. Hughes, Patrick Cadet, Peter L. Rady, Stephen K. Tyring, Robert Chin, and Eric M. Smith. Evidence for the Production and Action of Interleukin-10 in Pituitary Cells. Cellular and Molecular Neurobiology, Vol. 14, No. 1, 1994).

  •        In the discussion section, while it would be useful to look at how IL-10 polymorphism and level affects other types of cerebral tumors, it is too far stretched to describe its role in breast cancer, gastric cancer or head and neck cancer. Instead, the authors could add details about how polymorphisms in other interleukins/immune related genes are linked to PA.

Information was added:

Therefore study by Guo et al. suggest that IL-10 rs1800871 may be useful biomarkers for predicting glioma patient outcome (Hu M, Du J, Cui L, Huang T, Guo X, Zhao Y, Ma X, Jin T, Li G, Song J. IL-10 and PRKDC polymorphisms are associated with glioma patient survival. Oncotarget. 2016 Dec 6;7(49):80680-80687. doi: 10.18632/oncotarget.13028. PMID: 27811370; PMCID: PMC5348348.). Another study showed, that patients with primary central nervous system lymphomas (PCNSLs) and primary vitreoretinal lymphomas (PVRLs) patients had similar IL-10 (-1082) A allele frequencies, but genotype distributions differed from healthy controls. The findings suggest that the IL-10 (-1082) A allele is a risk factor for higher IL-10 levels in primary central nervous system lymphomas PVRLs and PCNSLs. Higher IL-10 levels have been correlated with more aggressive disease in both PVRLs and PCNSLs, making this finding an important and potentially clinically significant observation (Ramkumar HL, Shen DF, Tuo J, Braziel RM, Coupland SE, Smith JR, Chan CC. IL-10 -1082 SNP and IL-10 in primary CNS and vitreoretinal lymphomas. Graefes Arch Clin Exp Ophthalmol. 2012 Oct;250(10):1541-8. doi: 10.1007/s00417-012-2037-1. Epub 2012 May 25. PMID: 22628023; PMCID: PMC3469767.). Indeed is only one sudy analyzing other IL-9 SNP, done by Mickevicius et al.. This study proved that the G/G genotype and G allele of the IL-9 (rs1859430) polymorphism are more prevalent among patients with a recurrent PA. Moreover, our results show the that A/A genotype and A allele are significantly more common among those patients, whom PA did not reoccur. We can hypothesize that IL9 can be important for PA recurrence but it has no impact on hormonal activity or invasiveness. (Mickevicius T, Vilkeviciute A, Glebauskiene B, Kriauciuniene L, Liutkeviciene R. Do TRIB1 and IL-9 Gene Polymorphisms Impact the Development and Manifestation of Pituitary Adenoma? In Vivo. 2020 Sep-Oct;34(5):2499-2505. doi: 10.21873/invivo.12066. PMID: 32871778; PMCID: PMC7652474)

-        The methods and materials section is far too short. Even if some methods have been previously described in other studies, some description should still be provided for each of the used methods.

Thank you for your comment. Methods and materials section was updated, it was divided into subchapters and information added.

The methods and materials section should be divided into subchapters. There are some details missing. For instance, which ELISA kit was used?

Thank you for your comment. Methods and materials section was updated, it was divided into subchapters and information added.

IL-10 Human ELISA Kit was used.

-        The results section should also be divided into smaller subchapters, each one containing at the end a short summary of the findings.

Thank you for your comment. Results section was updated.

-        It could be interesting and more engaging to replace the way the results are presented. There are too many tables, and it gets difficult to follow the information. For instance, the ELISA results could be represented as graphs

Thank you for your comment. Results section was updated.

-        More details about inactive PA should be given, since the discovered polymorphisms are linked specifically to this type of PA

Thank you for your comment. Information was added.

The other group of authors also studied patients with nonfunctioning but invasive PA (NFPAs) and found lower numbers of CD3-CD56+ natural killer (NK) cells and significantly higher levels of CD3+CD8+CD28 T cells (CD8+ Tregs) and interleukin-10 (IL -10) in peripheral blood. In addition, patients with invasive NFPAs had fewer infiltrated CD56+ cells, fewer infiltrated CD28+ cells, and significantly higher IL -10 expression. These results indicated that low infiltration of CD56+ cells and CD28+ cells and high IL -10 expression in pituitary tumor tissues were related to increased invasiveness of NFPAs. The levels of CD3-CD56+ NK cells, CD8+ Tregs, and IL -10 in peripheral blood may be useful as diagnostic markers for invasive NFPAs (Huang X, Xu J, Wu Y, Sheng L, Li Y, Zha B, Sun T, Yang J, Zang S, Liu J. Alterations in CD8+ Tregs, CD56+ Natural Killer Cells and IL-10 Are Associated With Invasiveness of Nonfunctioning Pituitary Adenomas (NFPAs). Pathol Oncol Res. 2021 Mar 25;27:598887. doi: 10.3389/pore.2021.598887. PMID: 34257554; PMCID: PMC8262195.).

-        At the end of the discussion section, more details about how the results could be further analyzed or what it can mean for the study of PA should be given. For instance, the authors can include talks about personalized medicine and patients’ stratification or risk assessment in the general population

Information was added:

Personalised medicine plays the most important role in the prevention, diagnosis, prognosis, and therapy of neoplasms and cancer. Its importance for clinical management in routine clinical practise and treatment. Identification of genes predisposing to neoplasms and cancer can help to make decisions for individual disease risk modification. The study of genes in different populations is very valuable because the prevalence of genes varies widely in very different ethnic groups. In our future medicine, we need to focus on the role of personalised medicine, including the most important one - gene therapy medicine.

Reviewer 2 Report

Dear authors, 

I have some suggestions and questions:

Introduction: Please shorten the introduction and focus on relevant data for your study (for example, paragraphs explaninig pituitary adenoma diagnosis and treatment could be omitted).

Methods: 

You stated that you had two gorups of participants: first group comprised patients with pituitary adenoma, and healthy controls. my question is: how do you know that your healthy controls do not have putuitary adenoma?

Please remove the sentence on the second page, row number 93 and 94: "Therefore, the age of the patients was not statistically significant...." to the Results section.

Third page, rows 114 to 118: Please explain why IL-10 haplotype association analysis was performed in patients with multiple sclerosis and control goups.

Author Response

I have some suggestions and questions:

Dear Reviewer, thank you for your suggestions and questions. We have made some revisions in response to your comments.

Introduction: Please shorten the introduction and focus on relevant data for your study (for example, paragraphs explaninig pituitary adenoma diagnosis and treatment could be omitted).

Thank you for your comment. Introduction section was revised and updated.

Methods: 

You stated that you had two gorups of participants: first group comprised patients with pituitary adenoma, and healthy controls. my question is: how do you know that your healthy controls do not have putuitary adenoma?

The healthy controls group consisted of age and gender matched subjects having a good general health. They had no symptoms of the disease.

Please remove the sentence on the second page, row number 93 and 94: "Therefore, the age of the patients was not statistically significant...." to the Results section.

It was removed.

Third page, rows 114 to 118: Please explain why IL-10 haplotype association analysis was performed in patients with multiple sclerosis and control goups.

Sorry, there was a mistake. It was corrected.

Round 2

Reviewer 1 Report

The authors have significantly changed the manuscript. The articles is much improved.

Reviewer 2 Report

Thank you for the corrections! I think the manuscript could be accepted in its present form.